# The nutritional status of mycetoma affected patients seen at the Mycetoma Research Center, Sudan

**Manal Hassan Gabani[1], Arwa Abdelraouf Ahmed[1], Alshaima Abdelelah Hassan[1], Mona Abdelrahim Abdalla[1], Samar Abdelmahmoud Mustafa[1], Tasneem Abdelmutalab Alobaid[1], Abrar Adam Khatir[1], Reell Mukhlis Mohammed[1], Nehal Ibrahim Awad[1], Tanzeel Alqurashi Abdellateef[1], Abeer Hassan[1], Eiman Siddig Ahmed[1], Mohammed Zain Ali[2], Ahmed Hassan Fahal[1]\***

**1** Mycetoma Research Center, University of Khartoum, Khartoum, Sudan, **2** The School of Health Sciences, Ahfad University for Women, Omdurman, Sudan

\* ahfahal@mycetoma.edu.sd, ahfahal@gmail.com

**Data Availability Statement:** "All data are in the manuscript and Supporting information files".

**Funding:** The author(s) received no specific funding for this work.

## Abstract

Nutrition plays a critical and crucial role in addressing neglected tropical diseases (NTDs) and their complications, as they often contribute to malnutrition, which can worsen the impact of these conditions. Therefore, it is necessary to investigate the nutritional status of mycetoma patients, which has not been explored previously.

This descriptive cross-sectional hospital-based study was conducted at the Mycetoma Research Center (MRC), University of Khartoum, Sudan. The study included 179 confirmed mycetoma patients and an equal number of age- and sex-matched normal controls. The nutritional status of the mycetoma patients was assessed and compared with that of the control group. The majority of the patients were young adults with varying educational levels, predominantly from Central Sudan. The foot was the most commonly affected part; most patients had lesions more than 10 cm in diameter.

The Body Mass Index (BMI) was calculated for both study groups, revealing that 43.5% of the patients and 53.6% of controls had a normal BMI. Furthermore, 36% of patients were underweight, contrasting with only 11% in the control group. Correlation analyses indicated no significant associations between BMI and age groups, educational levels, daily meals, food quantity, and appetite in the study population ($p > 0.05$). Similarly, no significant differences were observed in BMI concerning disease duration and affected sites ($p = 0.0577$). The Kruskal-Wallis test did not reveal significant differences in BMI means among the groups.

The study revealed that most participants consumed three meals daily, and the control group showed a more robust appetite and consumed more food than the patient group ($p = 0.005$). Nevertheless, there were no significant differences in the consumption of different food types between the patient and control groups and among different BMI categories ($p = 0.025$ and $0.040$, respectively).

**Competing interests:** The authors have declared that no competing interests exist.

## Author summary

Mycetoma primarily affects low-income people in limited-resource communities in tropical and subtropical regions. The affected mycetoma are of low socioeconomic and health education status, and their localities' health and medical facilities are frequently inadequate. Hence, they present with massive, complicated disease; thus, the treatment outcome is suboptimal. Mycetoma has many disabilities and psychosocial effects, which may affect the patient nutritional status. Furthermore, most of the mycetoma epidemiological risk factors are indistinct and unclear. With this background, this study was conducted to determine the effect of the disease on the patients' nutritional status and to determine if nutrition has a role in mycetoma susceptibility. The study included 179 patients with confirmed mycetoma and age and sex-matched 179 controls from their communities. The mean patients' height was 160±14.31cm, and for the control was 166.78±10.97 cm. The patients' body weight mean was 56.09±17.67kg; for the control, it was 67.53±16.41kg, and no statistically significant differences existed. The body mass index showed a higher percentage of underweight individuals in the patients group, which was statistically significant. No significant correlations existed between the study population BMI groups and their demographic characteristics. There were many similarities between the studied patients and the control regarding the dietary habits and diet intake. More in-depth studies are needed to determine the causal and effect relationship between mycetoma and the affected population. Nutritional support and education, while encouraging intake of therapeutic/functional locally available food items, should be fundamental in the different mycetoma management activities and measurements.

## Introduction

Mycetoma is a serious, neglected tropical disease. The leading reason for the neglect is that it affects underprivileged populations in resource-constrained communities of the rural tropical and subtropical regions. It predominately affects individuals of low visibility and voice [1]. Furthermore, in endemic regions, the medical and health facilities are meagre and adequate treatment is lacking or inaccessible [2]. All these lead to a progressive disease with massive morbidities and can be fatal [3]. Mycetoma is a serious, neglected disease that continues to hinder and burden the poor, remote communities [4,5]. Mycetoma is a chronic disabling subcutaneous granulomatous inflammatory disease of fungal and bacterial origin [6]. The inflammatory granuloma progressively spreads to affect the skin, deep tissues and bones, leading to massive tissue damage, destruction and serious morbidities and can be fatal [7,8,9]. The triad of subcutaneous mass, multiple sinuses discharging purulent and sero-purulent discharge frequently containing grains is a disease characteristic [10,11]. The mycetoma occurrence and the risk factors association is still unclear. Likewise, the disease susceptibility, resistance, entry route, and incubation period are unclear [12,13,14].

Mycetoma is a disease of poor communities and remains a public health concern in underdeveloped countries [8,15,16,17]. It is ostracized by society for its destructive nature and its massive negative medical, health and socioeconomic consequences, including disability-related social stigma, poverty and psychosocial disturbances [8,18,19,20].

Presently, the diagnosis of mycetoma is difficult and tedious [21,22]. The available diagnostic tests are invasive, time-consuming, of low sensitivity and specificity, and there is no point of care [23,24,25,26,27,28]. The available treatment of mycetoma is suboptimal, characterised

by a low cure rate and high recurrence and follow-up rates [29,30]. The disease remains with patients for life, and a cure is rarely achieved.

Mycetoma typically affects young adults, the community breadwinners, leading to a vicious cycle of disease, disabilities, and poverty, and may culminate in malnutrition [8,31,32].

It is reported that nutrient supplementations and dietary lifestyle interventions are instrumental in reducing host oxidative stress, strengthening the immune system, mitigating comorbidities and are physiologically beneficial [33]. Nevertheless, in mycetoma, a significant knowledge gap exists concerning the effect of adequate nutrition on disease susceptibility, development, progress, and treatment outcome. With this background, the study set out to batch this gap to provide significant insight into the disease pathogenesis and treatment outcome and consolidate the existing body of literature.

## Patients and methods

### Ethics statement

The study population gave informed written consent, and the study ethical clearance was obtained from the Mycetoma Institutional Review Committee, MRC, No. MRC-IRB, 2021–77.

This descriptive cross-sectional hospital-based study was conducted at the Mycetoma Research Centre, University of Khartoum, Sudan. It included 179 patients with confirmed mycetoma and 179 age and sex-matched normal family members as a control.

The diagnosis of mycetoma was confirmed by meticulous clinical examinations, lesional ultrasound examination, histopathological examination of surgical biopsies and, grains culture and PCR examinations. Confirmed patients and controls were selected with a simple convenience sampling technique. Each patient was paired with control from their families, and age and sex were matched. Institutionalised persons, including prisons, dormitories, nursing homes, and patients with wounds from causes other than mycetoma, were excluded.

An expert clinical nutritionist and eight well-trained medical doctors interviewed the study population. The interview included their demographic characteristics and nutritional habits. They had their weights and heights measured, and their BMI was determined. A pre-designed, validated digital interview-based questionnaire was used for data collection.

Study participants were classified into five groups based on the World Health Organization (WHO) BMI scale classification [34]. The BMI of amputated patients was reported and considered in the data analysis.

The Statistical Packages for Social Sciences (SPSS) programme (version 26, USA) was used to manage the data. Appropriate statistical tests were used for data quantitative analysis, including the paired t-test to compare the nutritional status of the patients and controls. Statistical significance differences (p-value, Odds ratio) were calculated.

## Results

The study included 179 patients with confirmed mycetoma; 105 patients (58.7%) were males and 74 (41.3%) were females. The control group included 136(76%) males, and 43(24%) females. Most patients and controls (55.9%) were young adults in the age group 21–40 years with a mean age of 34.7±17.8 years and 36.68±14.31 years, respectively. The study population has different education levels, but most (47.5%) had primary education. Most of the study population was from the Gezira and Khartoum States (51.3%). The studied patients were mostly housewives (26%), students (16.6%), and unemployed (21.5%). In the control group, most of them were students, housewives, 12% each and sellers (9.4%), Table 1. The disease duration ranged from a few months to several years, with a mean of 8.35±8.40 years.

**Table 1. The study population Socio-demographic characteristics.**

| | Patients | | Control | |
|---|---|---|---|---|
| | **No.** | **%** | **No.** | **%** |
| **Gander** | | | | |
| Male | 105 | 58.7 | 136 | 76.0 |
| Female | 74 | 41.3 | 43 | 24.0 |
| **Age Group (years)** | | | | |
| <20 | 47 | 26.3 | 17 | 9.5 |
| 21–40 | 78 | 43.6 | 100 | 55.9 |
| >41 | 54 | 30.2 | 62 | 34.6 |
| **Education Level** | | | | |
| Illiterate | 36 | 20.1 | 20 | 11.2 |
| Quoranic/Khalwa | 21 | 11.7 | 11 | 6.1 |
| Primary-School | 85 | 47.5 | 44 | 24.6 |
| Post-Graduate | 27 | 15.1 | 54 | 30.2 |
| University | 08 | 4.5 | 41 | 22.9 |
| | 02 | 1.1 | 09 | 5.0 |
| **Residence** | | | | |
| Blue-Nile | 01 | 0.6% | 02 | 1.1% |
| East-Darfur | 02 | 01.1 | 02 | 01.1 |
| Gedarif | 03 | 01.7 | 04 | 02.2 |
| Gezira | 52 | 28.7 | 58 | 32.0 |
| Kassala | 03 | 01.7 | 05 | 02.8 |
| Khartoum | 40 | 22.1 | 21 | 11.6 |
| North-Darfur | 03 | 01.7 | 05 | 02.8 |
| North-Kordofan | 17 | 09.4 | 17 | 09.4 |
| Northern | 02 | 01.1 | 01 | 00.6 |
| Red-Sea | 01 | 0.6 | 03 | 01.0 |
| River-Nile | 07 | 03.9 | 06 | 03.3 |
| Sennar | 12 | 06.6 | 16 | 08.8 |
| South-Darfur | 05 | 02.8 | 05 | 02.8 |
| South Kordofan | 00 | 00.0 | 01 | 00.6 |
| West Darfur | 03 | 01.7 | 03 | 01.7 |
| West Kordofan | 01 | 00.6 | 01 | 00.6 |
| White-Nile | 29 | 16.0 | 31 | 17.1 |
| **Occupation** | | | | |
| Farmer | 21 | 11.6 | 14 | 07.70 |
| Shepherd | 6 | 03.3 | 2 | 01.10 |
| Worker | 8 | 04.4 | 18 | 09.90 |
| Seller | 5 | 02.8 | 17 | 09.40 |
| Student | 30 | 16.6 | 22 | 12.20 |
| House Wife | 47 | 26.0 | 22 | 12.20 |
| Unemployment | 39 | 21.5 | 15 | 08.30 |
| Other | 23 | 02.7 | 71 | 39.20 |

**Table 2. The mycetoma site and size distributions.**

| Site | No. | % |
|---|---|---|
| Foot | 118 | 65.2 |
| Hand | 19 | 10.5 |
| Knee | 11 | 6.1 |
| Leg | 07 | 3.9 |
| Gluteal | 03 | 1.7 |
| Heal | 03 | 1.7 |
| Hip | 02 | 1.1 |
| Head | 02 | 1.1 |
| Thigh | 02 | 1.1 |
| Testis | 01 | 0.6 |
| Abdominal | 01 | 0.6 |
| Back | 01 | 0.6 |
| Buttock | 01 | 0.6 |
| Foot & Knee | 01 | 0.6 |
| Foot & Leg | 01 | 0.6 |
| Knee & Leg | 01 | 0.6 |
| Rectal | 01 | 0.6 |
| **Lesion Size** | **No.** | **%** |
| Small < 5cm | 19 | 10.6 |
| Medium 5 -10cm | 42 | 23.5 |
| Massive >10cm | 118 | 65.9 |

The foot (65.2%), hand (10.5%), knee (6.1%) and leg (3.9%) were affected most in this study. The majority of the studied patients, 118(65.9%), had massive lesions more than 10 cm in diameter, and 42 patients (23.5%) had moderate lesions (5–10 cm in diameter), Table 2.

## Body mass index

The mean patients' height was 160±14.31cm, and for the control, it was 166.78±10.97cm. The patients' body weight mean was 56.09±17.67kg; for the control, it was 67.53±16.41kg. The body mass index for the study population was calculated, revealing that 43.5% of the patients and 53.6% of the control had normal weight. It showed 36% of the patients and 11% of the control were underweight. Table 3

There are no significant correlations between the study population BMI groups and their age groups, educational level, meals taken per day, amount of food taken and their appetite, $p > 0.05$ and the disease duration and sites, $p = 0.0577$. Kruskal Wallis test revealed no significant difference between the means of BMI groups. (Results shown in S1).

**Table 3. The distribution of the study population according to the BMI.**

| Group | BMI (kg/m2) | Patients No. (%) | Control No. (%) |
|---|---|---|---|
| Underweight | <18.50 | 64 (35.7%) | 19 (11%) |
| Normal Weight | 18.5–24.9 | 78(43.5%) | 96 (53.6%) |
| Overweight | 25.0–29.9 | 25 (13.9%) | 46 (25.6%) |
| Obese I | 30.0–34.9 | 08 (4.4%) | 14 (7.1%) |
| Obese II | ≥35.0 | 04 (2.2%) | 04 (2.2%) |

## The dietary intake

The study showed most of the study population took three meals per day. The control group had a better appetite and ate more food than the patients group, $p = 0.005$. The study showed that 38% of the patients and 48% of the control group took meat daily. Regarding cheese intake, most of the study population did not take it daily, and 30% of the patients and control took rice 2 to 3 times per week. Less than 20% of patients and control did not take lemon daily. Less than 20% of patients took fava-beans daily, whereas more than 20% of control did, Table 4.

The dietary intake and habits distribution according to the BMI classification are shown in (Results shown in S1).

The study revealed no significant differences in cereal consumption between patients and the control group, as well as among different BMI groups ($p = 0.025$ and $0.040$, respectively). Sunflower oil intake was significantly associated with the BMI of the control group ($p = 0.025$), while meat and eggs showed significant associations with the control group's BMI ($p = 0.044$ and $0.040$, respectively). No significant association was observed between dairy product consumption and population BMI. Pigeon peas in the legumes group correlated significantly with both patients and control groups, as well as BMI groups ($p = 0.004$ and $0.025$, respectively). Fruits showed insignificant correlations with patients and control. In the drinks and juices group, bottled drinks significantly correlated with the BMI of the control group ($p = 0.015$). Cooked vegetables had significant associations with both control and patient groups and the BMI ($p = 0.005$ and $0.018$, respectively).

There were no significant differences in the consumption of various cereal types between the patients and control groups, as well as among different BMI groups, $p = 0.025$ and $0.040$, respectively. Regarding oil-rich foods, a higher intake of sunflower oil was significantly associated with the BMI of the control group (p = 0.025). Among animal products, only meat and eggs showed significant associations with the BMI of the control group ($p = 0.044$ and $0.040$, respectively), while there was no significant association between dairy product consumption and population BMI. In the legumes group, the consumption of pigeon peas significantly correlated with both patients and control groups, as well as BMI groups, $p = 0.004$ and $0.025$, respectively. All types of fruits consumed in the study showed insignificant correlations with patients and control BMIs. In the drinks and juices food group, the consumption of bottled drinks significantly correlated with the BMI of the control group ($p = 0.015$). The consumption of cooked vegetables had significant associations with both control and patients groups, as well as BMI ($p = 0.005$ and $0.018$, respectively).

**Table 4. The study population's dietary habits.**

| Item | Ranges | Control No. (%) | | Patients No. (%) | |
|---|---|---|---|---|---|
| Meals taken per day | 1 | 02 | 01.1% | 00 | 00.0% |
| | 2 | 70 | 39.1% | 79 | 44.2% |
| | 3 | 107 | 59.7% | 100 | 55.8% |
| Food amount taken | Less | 27 | 15.0% | 44 | 24.3% |
| | Same | 121 | 67.5% | 123 | 68.0% |
| | More | 31 | 17.3% | 14 | 07.7% |
| Appetite | Low | 26 | 14.5% | 56 | 31.2% |
| | Moderate | 120 | 67.0% | 109 | 60.8% |
| | High | 33 | 18.4% | 14 | 07.8% |

## Discussion

Several NTDs, such as intestinal helminthiasis and schistosomiasis, frequently induce serious nutrients malabsorption and malnutrition [35]. These infections can lead to malabsorption of many essential vitamins, minerals and other elements from the diet [36]. Furthermore, malnutrition can increase the susceptibility to many NTDs and other infections. Thus, adequate nutrition is indispensable for maintaining dynamic immune responses to defend the human body against such infections [37]. Furthermore, proper nutrition is vital to support the affected patients in dealing with the NTDs and minimise the long-term NTDs complications, and mycetoma is not an exception.

In this study, 35.4% of the studied patients were underweight, 43.5% had normal weight, 13.8% were overweight, and 7.8% were obese. The incidence of being underweight is higher compared to the control (10.5%). However, it is difficult to determine the causal and effect relationship in this group. The low weight can be disease-induced, or it can be a disease risk factor, and this demands more investigations. The mycetoma patients' weight distributions reported in this study differ from the weight of those with leprosy, a mycetoma similar disease. One report from Brazil showed that 60% of leprosy patients had a normal BMI, 14% were underweight, and 26% were overweight or obese [38]. The cause is unclear, but the studied populations probably lacked homogeneity. However, maintaining a healthy body weight is vital, as excessive weight can strain the mycetoma-affected areas and potentially worsen deformities and disabilities.

This study showed no clear association between the patients' BMI and mycetoma. The literature revealed contraindicating reports on the effects of nutrition on other NTDs. In leprosy, many studies reported normal BMI in most patients, and in some studies, a high incidence of obesity (60%) was reported [39,40]. These studies showed no association between leprosy reactional states and BMI. Furthermore, BMI failed to correlate with the presence of reaction leprosy and deformities or multibacillary disease in the reported studies ($p>0.05$) [41,42]. Another study from Indonesia reported no significant association between poor nutrition and leprosy ($p = 0.008$, OR = 0.422) [43]. However, a substantial association between undernutrition (BMI<18.5) and leprosy and WHO grade 2 disability was reported [44]. One study reported a higher incidence of growth disorders and anaemia in children with tuberculous meningitis complicated with pulmonary and abdominal tuberculosis. In these patients, the nutritional status was one of the causes of abandoning treatment [45].

In many NTDs, food and its availability are a substantial prevention and management influence [46]. Most of the NTDs affected patients from low socioeconomic strata have lifelong food insecurity, with enormous psychosocial and physical disabilities to access the available meagre food [47]. It is clear nutrition influences the patients' immunity, but the mechanism is unclear. Furthermore, the exact role of nutrition on susceptibility and development of mycetoma and other NTDs is still indistinct and complicated.

In leprosy, case-control studies on socioeconomic factors showed that the disease is sometimes associated with food shortage [48,49]. Inadequate food intake leads to reduced intake of carbohydrates, proteins, fats, vitamins, and minerals, and this nutritional deficiency impairs the immune system against several infections [50]. However, in mycetoma, that is unclear. As reported in Brazil, the socioeconomic policies and population health conditions in developing nations are closely related and can influence the NTDs new case detection. The high coverage of the Brazilian Conditional Cash Transfer and Primary Health Care programmes significantly reduced the leprosy new case detection; however, the underlying mechanism between food, poverty, and leprosy remains unclear [51].

The various diet types contain different nutritional elements such as protein, fat, carbohydrates, vitamins, minerals, trace elements, phytochemicals, probiotics and antioxidants. All have different roles in supporting body functions during stressful events, particularly chronic infections such as mycetoma. [52]. This study documented that the food type, the study population takes is the normal Sudanese diet, and more in-depth studies are needed to quantify that and see if some nutritional elements are deficient, if any, and their effects on the disease progress and treatment outcomes.

Nutrient supplementation has been influential in suppressing the host oxidative stress, strengthening immune responsiveness and mitigating potential adverse events in leprosy and diabetes [53]. Furthermore, the implications of vitamin and mineral supplementation in leprosy have demonstrated a significant reduction in symptoms and therapeutic requirements [54,55,56]. As mycetoma shares many characteristics of leprosy, it is worth trying nutrient supplementations for these patients to improve the disease progress and treatment outcome.

In this study, only clinical assessment was used to determine the nutritional status of the studied mycetoma patients, which is a limitation. Further studies, including biochemical nutritional parameters and body composition by bioelectrical impedance analysis, are needed for better insight into the effects of patients' nutritional status on the disease's clinical progress and treatment outcome. The successful intake of probiotics as ingestible adjuvants and immune modulators for antiviral immunity and management of SARS-CoV-2 infection and COVID-19 is worth studying in mycetoma.[57].

In conclusion, nutritional assessment and education should be integral parts of the mycetoma management guidelines. Patients should have optimal nutritional status, which is essential for appropriately functioning immune response, adequate response to mycetoma chronic infection and postoperative wound healing. Further in-depth research work is needed to determine the nutritional elements' effects on mycetoma pathogenesis and the treatment outcome.

## Supporting information

**S1 Table. The dietary intake and habits of the study population according to the BMI classification.**
(DOCX)

**S2 Table. Food items consumed by the study population by Body Mass Index classification.**
(DOCX)

## Author Contributions

**Conceptualization:** Manal Hassan Gabani, Arwa Abdelraouf Ahmed, Alshaima Abdelelah Hassan, Mona Abdelrahim Abdalla, Samar Abdelmahmoud Mustafa, Tasneem Abdelmutalab Alobaid, Abrar Adam Khatir, Reell Mukhlis Mohammed, Nehal Ibrahim Awad, Tanzeel Alqurashi Abdellateef, Abeer Hassan, Eiman Siddig Ahmed, Mohammed Zain Ali, Ahmed Hassan Fahal.

**Data curation:** Manal Hassan Gabani, Arwa Abdelraouf Ahmed, Alshaima Abdelelah Hassan, Mona Abdelrahim Abdalla, Samar Abdelmahmoud Mustafa, Tasneem Abdelmutalab Alobaid, Abrar Adam Khatir, Reell Mukhlis Mohammed, Nehal Ibrahim Awad, Tanzeel Alqurashi Abdellateef, Abeer Hassan, Eiman Siddig Ahmed, Mohammed Zain Ali, Ahmed Hassan Fahal.

**Formal analysis:** Manal Hassan Gabani, Arwa Abdelraouf Ahmed, Alshaima Abdelelah Hassan, Mona Abdelrahim Abdalla, Samar Abdelmahmoud Mustafa, Tasneem Abdelmutalab

Alobaid, Abrar Adam Khatir, Reell Mukhlis Mohammed, Nehal Ibrahim Awad, Tanzeel Alqurashi Abdellateef, Abeer Hassan, Eiman Siddig Ahmed, Mohammed Zain Ali, Ahmed Hassan Fahal.

**Investigation:** Manal Hassan Gabani, Arwa Abdelraouf Ahmed, Alshaima Abdelelah Hassan, Samar Abdelmahmoud Mustafa, Tasneem Abdelmutalab Alobaid, Abrar Adam Khatir, Reell Mukhlis Mohammed, Nehal Ibrahim Awad, Tanzeel Alqurashi Abdellateef, Abeer Hassan, Eiman Siddig Ahmed, Mohammed Zain Ali, Ahmed Hassan Fahal.

**Methodology:** Manal Hassan Gabani, Arwa Abdelraouf Ahmed, Alshaima Abdelelah Hassan, Mona Abdelrahim Abdalla, Samar Abdelmahmoud Mustafa, Tasneem Abdelmutalab Alobaid, Abrar Adam Khatir, Reell Mukhlis Mohammed, Nehal Ibrahim Awad, Tanzeel Alqurashi Abdellateef, Abeer Hassan, Eiman Siddig Ahmed, Mohammed Zain Ali, Ahmed Hassan Fahal.

**Project administration:** Manal Hassan Gabani, Mohammed Zain Ali.

**Supervision:** Eiman Siddig Ahmed, Mohammed Zain Ali, Ahmed Hassan Fahal.

**Validation:** Mohammed Zain Ali.

**Visualization:** Manal Hassan Gabani, Arwa Abdelraouf Ahmed, Alshaima Abdelelah Hassan, Mona Abdelrahim Abdalla, Samar Abdelmahmoud Mustafa, Tasneem Abdelmutalab Alobaid, Abrar Adam Khatir, Reell Mukhlis Mohammed, Nehal Ibrahim Awad, Tanzeel Alqurashi Abdellateef, Abeer Hassan, Eiman Siddig Ahmed, Mohammed Zain Ali, Ahmed Hassan Fahal.

**Writing – original draft:** Manal Hassan Gabani, Arwa Abdelraouf Ahmed, Alshaima Abdelelah Hassan, Mona Abdelrahim Abdalla, Samar Abdelmahmoud Mustafa, Tasneem Abdelmutalab Alobaid, Abrar Adam Khatir, Reell Mukhlis Mohammed, Nehal Ibrahim Awad, Tanzeel Alqurashi Abdellateef, Abeer Hassan, Eiman Siddig Ahmed, Mohammed Zain Ali, Ahmed Hassan Fahal.

**Writing – review & editing:** Manal Hassan Gabani, Arwa Abdelraouf Ahmed, Alshaima Abdelelah Hassan, Mona Abdelrahim Abdalla, Samar Abdelmahmoud Mustafa, Tasneem Abdelmutalab Alobaid, Abrar Adam Khatir, Reell Mukhlis Mohammed, Nehal Ibrahim Awad, Tanzeel Alqurashi Abdellateef, Abeer Hassan, Eiman Siddig Ahmed, Mohammed Zain Ali, Ahmed Hassan Fahal.

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
