## [Decision Letter · Decision Letter 0]

14 Nov 2023

Dear Prof. Fahal,

Thank you very much for submitting your manuscript "The Nutritional Status of Mycetoma Affected Patients seen at the Mycetoma Research Center, Sudan." for consideration at PLOS Neglected Tropical Diseases. As with all papers reviewed by the journal, your manuscript was reviewed by members of the editorial board and by several independent reviewers. In light of the reviews (below this email), we would like to invite the resubmission of a significantly-revised version that takes into account the reviewers' comments. 

We cannot make any decision about publication until we have seen the revised manuscript and your response to the reviewers' comments. Your revised manuscript is also likely to be sent to reviewers for further evaluation.

Sincerely,

Joshua Nosanchuk, MD

Section Editor

Reviewer's Responses to Questions

**Key Review Criteria Required for Acceptance?**

**Methods**

-Are the objectives of the study clearly articulated with a clear testable hypothesis stated?

-Is the study design appropriate to address the stated objectives?

-Is the population clearly described and appropriate for the hypothesis being tested?

-Is the sample size sufficient to ensure adequate power to address the hypothesis being tested?

-Were correct statistical analysis used to support conclusions?

-Are there concerns about ethical or regulatory requirements being met?

Reviewer #1: The objectives are clear. But I have some queries. How was the family member from the control group chosen? Were the controls paired by age? In case no family member was present, how was that control chosen? Was that patient excluded from the analysis if no family member was available? How was the nutritional interview performed? Was a specific format or validated questionnaire employed? Please include this in the methods section. Please include in the methods section how the “appetite” was determined.

**Results**

-Does the analysis presented match the analysis plan?

-Are the results clearly and completely presented?

-Are the figures (Tables, Images) of sufficient quality for clarity?

Reviewer #1: In the results section, please include kg and cm where appropriate. I believe the last paragraph of the result section could be improved. I don’t think it is necessary to present some foods such as cucumber, carrot, and sunflower oil consumption as independent variables. I believe it would be better to group certain food groups and analyze them together (example: vegetables, fruits, etc).

**Conclusions**

-Are the conclusions supported by the data presented?

-Are the limitations of analysis clearly described?

-Do the authors discuss how these data can be helpful to advance our understanding of the topic under study?

-Is public health relevance addressed?

Reviewer #1: The discussion is well-presented and written. The limitations are clearly stated (I would add that body composition by bioelectrical impedance analysis is needed too) and the conclusion is valid. I would modify this sentence: “The successful intake of probiotics as ingestible adjuvants and immune modulators for antiviral immunity and management of SARS-CoV-2 infection and COVID-19 is worth studying in mycetoma.” as the evidence of their usefulness in COVID-19 is poor.

**Editorial and Data Presentation Modifications?**

Reviewer #1: Nil.

**Summary and General Comments**

Reviewer #1: The authors present an interesting study of nutrition in patients with mycetoma from Sudan. The methods section could be improved by thoroughly explaining the control group and the nutritional questionnaire employed. The article is generally well-written and information about nutritional status in patients with mycetoma is very scarce.

PLOS authors have the option to publish the peer review history of their article (what does this mean?). If published, this will include your full peer review and any attached files.

Reviewer #1: No
---

## [Decision Letter · Decision Letter 1]

30 Nov 2023

Dear Prof. Fahal,

Thank you very much for submitting your manuscript "The Nutritional Status of Mycetoma Affected Patients seen at the Mycetoma Research Center, Sudan." for consideration at PLOS Neglected Tropical Diseases. As with all papers reviewed by the journal, your manuscript was reviewed by members of the editorial board and by several independent reviewers. The reviewers appreciated the attention to an important topic. Based on the reviews, we are likely to accept this manuscript for publication, providing that you modify the manuscript according to the review recommendations. 

Sincerely,

Joshua Nosanchuk, MD

Section Editor

Reviewer's Responses to Questions

**Key Review Criteria Required for Acceptance?**

**Methods**

-Are the objectives of the study clearly articulated with a clear testable hypothesis stated?

-Is the study design appropriate to address the stated objectives?

-Is the population clearly described and appropriate for the hypothesis being tested?

-Is the sample size sufficient to ensure adequate power to address the hypothesis being tested?

-Were correct statistical analysis used to support conclusions?

-Are there concerns about ethical or regulatory requirements being met?

Reviewer #1: Methods have improved and queries have been addressed.

**Results**

-Does the analysis presented match the analysis plan?

-Are the results clearly and completely presented?

-Are the figures (Tables, Images) of sufficient quality for clarity?

Reviewer #1: Some suggestions are included below.

**Conclusions**

-Are the conclusions supported by the data presented?

-Are the limitations of analysis clearly described?

-Do the authors discuss how these data can be helpful to advance our understanding of the topic under study?

-Is public health relevance addressed?

Reviewer #1: Conclusions are adequate.

**Editorial and Data Presentation Modifications?**

Reviewer #1: See below.

**Summary and General Comments**

Reviewer #1: The authors have improved the manuscript. Some corrections are still needed.

The abstract needs extensive modification. The authors should include, not only results, but some of the conclusions that are present in the main manuscript. Please improve the abstract by including the most relevant results of your analysis. 

Sentence “The Body Mass Index (BMI) for the study population was calculated, revealing that 43.3% of the patients and 50.3% of the control had BMI.” is incorrect. Please correct it.

The previous query “I don’t think it is necessary to present some foods such as cucumber, carrot, and sunflower oil consumption as independent variables. I believe it would be better to group certain food groups and analyze them together (example: vegetables, fruits, etc)” was responded by the authors as:

“There is a lot of data obtained, and we found it difficult not to show that. “

Please justify including variables like “cucumber and carrot consumption” as independent variables and include a justification for this and its relevance in the discussion.

PLOS authors have the option to publish the peer review history of their article (what does this mean?). If published, this will include your full peer review and any attached files.

Reviewer #1: No

Figure Files:

Data Requirements:

Reproducibility:

References

---

## [Editor Report · Decision Letter 2]

11 Dec 2023

Dear Prof. Fahal,

Thank you very much for submitting your manuscript "The Nutritional Status of Mycetoma Affected Patients seen at the Mycetoma Research Center, Sudan." for consideration at PLOS Neglected Tropical Diseases. As with all papers reviewed by the journal, your manuscript was reviewed by members of the editorial board and staff. The reviewers appreciated the attention to an important topic. 

In your response to the question on BMI, you stated "it is correct." However, it is not. Here are the sentences:

"The body mass index for the study population was calculated, revealing that 43.3% of the patients and 50.3% of the control had normal weight. It showed 36% of the patients and 7% of the control were underweight. Table 3"

Yet, the table instead shows that it is 43.1% and not 43.3% for normal weigh in patients & 53% not 50.3% for controls with normal weight. For underweight, the numbers are also different. Your table is shown below. 

Group BMI (kg/m2) Patients

No. (%) Control

No. (%)

Underweight <18.50 64 (35.4%) 19 (10.5%)

Normal Weight 18.5-24.9 78(43.1%) 96 (53.0%)

Overweight 25.0-29.9 25 (13.8%) 46 (25.4%)

Obese I 30.0-34.9 09 (5.0%) 15 (8.3%)

Obese II ≥35.0 05 (2.8%) 05 (2.8%)

The editorial board asks that you carefully review your calculations across the manuscript and specifically indicate in your next response letter that you have had a statistician confirm your data.

Also, please put "Table 3" into the prior sentence and close it with a period. 

Sincerely,

Joshua Nosanchuk, MD

Section Editor

Figure Files:

Data Requirements:

Reproducibility:

References

---

## [Editor Report · Decision Letter 3]

14 Dec 2023

Dear Prof. Fahal,

Thank you very much for submitting your manuscript "The Nutritional Status of Mycetoma Affected Patients seen at the Mycetoma Research Center, Sudan." for consideration at PLOS Neglected Tropical Diseases. 

The journal staff note that the authors corrected the numbers for the individuals with normal BMI, but did not correct those in the control group. In line 35 and 175, the authors report 7% of controls are underweight, yet the table, line 184, shows that the number is 10.6% (or 11% rounded). Please correct this and review all calculations in the manuscript carefully.

Sincerely,

Joshua Nosanchuk, MD

Section Editor

Figure Files:

Data Requirements:

Reproducibility:

References

---

## [Editor Report · Decision Letter 4]

18 Dec 2023

Dear Prof. Fahal,

We are pleased to inform you that your manuscript 'The Nutritional Status of Mycetoma Affected Patients seen at the Mycetoma Research Center, Sudan.' has been provisionally accepted for publication in PLOS Neglected Tropical Diseases.

Best regards,

Joshua Nosanchuk, MD

Section Editor

---

## [Editor Report · Acceptance letter]

27 Dec 2023

Dear Prof. Fahal,

We are delighted to inform you that your manuscript, "The Nutritional Status of Mycetoma Affected Patients seen at the Mycetoma Research Center, Sudan.," has been formally accepted for publication in PLOS Neglected Tropical Diseases.

Best regards,

Shaden Kamhawi

co-Editor-in-Chief

Paul Brindley

co-Editor-in-Chief
